



# Summertime aerosol volatility measurements in Beijing, China

Weiqi Xu[1,2], Conghui Xie[1,2], Eleni Karnezi[3,a], Qi Zhang[4], Junfeng Wang[5], Spyros N. Pandis[3], Xinlei Ge[5],

Qingqing Wang[1], Jian Zhao[1,2], Wei Du[1,2,b], Yanmei Qiu[1,2], Wei Zhou[1,2], Yao He[1,2], Jingwei Zhang[1,2],

Junling An[1,2], Ying Li[1], Jie Li[1], Pingqing Fu[2,6], Zifa Wang[1,2], Douglas R. Worsnop[7], and Yele Sun[1,2,8*]

[1]State Key Laboratory of Atmospheric Boundary Layer Physics and Atmospheric Chemistry, Institute of Atmospheric Physics, Chinese Academy of Sciences, Beijing 100029, China
[2]University of Chinese Academy of Sciences, Beijing 100049, China
[3]Department of Chemical Engineering, Carnegie Mellon University, Pittsburgh, PA, USA
[4]Department of Environmental Toxicology, University of California, 1 Shields Ave., Davis, California 95616, United States
[5]School of Environmental Science and Engineering, Nanjing University of Information Science & Technology, Nanjing 210044, China
[6]Institute of Surface-Earth System Science, Tianjin University, Tianjin 300072, China
[7]Aerodyne Research Inc., Billerica, Massachusetts 01821, USA
[8]Center for Excellence in Regional Atmospheric Environment, Institute of Urban Environment, Chinese Academy of Sciences, Xiamen 361021, China
[a]now at: Earth Sciences Department, Barcelona Supercomputing Center, BSC-CNS, Barcelona 08034, Spain
[b]now at: Department of Physics, University of Helsinki, P.O. Box 64, 00014 Finland

*Correspondence*: Yele Sun (sunyele@mail.iap.ac.cn)

**Abstract.** Volatility plays a key role in affecting mass concentrations and lifetime of aerosol particles in the atmosphere, yet our knowledge of aerosol volatility in relatively polluted environment, e.g., north China remains poor. Here aerosol volatility in Beijing in summer 2017 and 2018 was measured using a thermodenuder (TD) coupled with an Aerodyne high-resolution aerosol mass spectrometer (AMS) and a soot particle AMS. Our results showed overall similar thermograms for most non-refractory aerosol species compared with those reported in previous studies. However, high mass fraction remaining and $NO^+/NO_2^+$ ratio for chloride and nitrate, respectively above 200 °C indicated the presence of considerable metallic salts and organic nitrates in Beijing. The volatility distributions of organic aerosol (OA) and four OA factors that were resolved from positive matrix factorization were estimated using a mass transfer model. The ambient OA comprised mainly semi-volatile organic compounds (SVOC, 63%) with an average effective saturation concentration ($C^*$) of 0.55 µg m$^{-3}$, suggesting overall more volatile properties than OA in megacities of Europe and US. Further analysis showed that the freshly oxidized secondary OA (LO-OOA) was the most volatile OA factor (SVOC = 70%) followed by hydrocarbon-like OA (HOA). In contrast, the volatility of more oxidized SOA (MO-OOA) was comparable to that of cooking OA with SVOC on average accounting for 60.2%. We also compared the volatility of ambient and black carbon–containing OA. Our results showed that the BC-containing primary OA (POA) was much more volatile than ambient POA ($C^*$= 0.69 µg m$^{-3}$ vs. 0.37 µg m$^{-3}$), while the BC-containing SOA was much less volatile, highlighting the very different composition and properties between BC-containing and ambient aerosol particles.



# 1 Introduction

Atmospheric aerosols can cause a series of health risks (Lelieveld et al., 2015) and affect the earth's radiative balance (Boucher et al., 2013). As one of the most important properties, volatility modulates mass concentrations and size distributions of aerosol particles via gas-particle partitioning, and hence influences hygroscopicity, optical properties, and

fate of related compounds (Topping and McFiggans, 2012;Donahue et al., 2012). Traditionally, "two-product model" (Odum et al., 1996) has been used to parameterize the volatility distribution of secondary organic aerosol (SOA), yet it often underestimates ambient SOA substantially (Li et al., 2013;Heald et al., 2005). Donahue et al. (2006) updated the volatility distribution framework using the "Volatility Basis Set" (VBS) consisting of logarithmically-spaced effective saturation concentration ($C^*$) bins over a wide range which improves the model simulations of SOA significantly. However, there is

still a large model-observation gap in predicting atmospheric organic aerosol (Zhang et al., 2013;Tsigaridis et al., 2014). One reason is our incomplete understanding of organic aerosol (OA) volatility in various environments.

The thermodenuder (TD) coupled with Aerodyne aerosol mass spectrometer (AMS) has been widely used to measure chemically-resolved aerosol volatility in field campaigns (Huffman et al., 2009a;Huffman et al., 2009b) and laboratory studies (Kolesar et al., 2015;Saha and Grieshop, 2016). The mass or volume fraction remaining (MFR/VFR), a ratio of the

mass/volume of the aerosol remaining after passing through a heated section to the species mass/volume without heating, is often used as an indicator of volatility, and larger MFR indicates lower volatility (Huffman et al., 2009a;An et al., 2007). For example, Huffman et al. (2009b) found that both ambient primary OA (POA) and SOA showed semi-volatile properties that contradicted with the representation of OA volatility in most traditional models. MFR is also affected by the enthalpy of vaporization, initial concentration, residence time in heated section, aerosol size distribution, and potential mass transfer

resistances (Saleh et al., 2011), therefore, it may lead to erroneous conclusions using MFR only as an indicator of volatility. For example, Kostenidou et al. (2018) found that SOA species with higher MFR can be more volatile because of lower enthalpy of vaporization. As a result, a mass transfer model taking into account during the dynamic evaporation of the aerosol all these properties that affect volatility as vaporization enthalpy residence time, particle size and OA concentration into account is needed for better interpretation of OA volatility measurements (Riipinen et al., 2010).

A number of studies have been conducted to investigate the OA volatility using thermogram models assuming fixed effective vaporization enthalpy and mass accommodation coefficient (Cappa and Jimenez, 2010;Lee et al., 2010;Paciga et al., 2016;Louvaris et al., 2017;Kostenidou et al., 2018). The results showed that OA volatility distributions may vary from place to place, and the estimated OA volatility was sensitive to the assumed values of the effective vaporization enthalpy and the mass accommodation coefficient (Riipinen et al., 2010). Saha et al. (2015) used a "dual thermodenuder" system to better

constrain the estimated values by varying both temperature and residence time. Karnezi et al. (2014) proposed an improved





experimental approach combining TD and isothermal dilution measurements and introduced a method for the estimation and the uncertainty range for the estimated volatility distribution together with the vaporization enthalpy and accommodation coefficient. Aerosol volatility can also be estimated with a semi-empirical approach from the gas and particle phase measurements of molecules using chemical ionization mass spectrometer equipped with a Filter Inlet for Gases and

AEROsols (FIGAERO–CIMS). Recently, Stark et al. (2017) evaluated the volatility distributions of OA from three different methods, and found that the thermogram method from TD-AMS measurements could be the best for quantification of aerosol volatility distributions.

Despite this, few volatility measurements have been reported in China, especially in northern China with high concentrations of $PM_{2.5}$ (Sun et al., 2015;Li et al., 2017). Bi et al. (2015) measured the volatility of individual aerosol particles in the Pearl

River Delta (PRD) region using a single particle AMS coupled with a TD. The results showed that the volatility of elemental carbon (EC)-containing particles may depend on particle types and molecular formulas of secondary ions. Cao et al. (2018) investigated aerosol volatility in winter in PRD region using a TD-AMS system. The results of MFR showed that hydrocarbon-like OA (HOA) was the most volatile OA component followed by less oxidized OOA (LO-OOA), cooking and biomass burning OA (BBOA), and more oxidized OOA (MO-OOA). However, aerosol volatility in different seasons and

different regions in China remains poorly understood.

In this study, aerosol volatility was measured using a TD coupled with a high-resolution AMS (TD-HR-AMS) and soot particle AMS (TD-SP-AMS) in summer in 2018 and 2017 in Beijing. The OA composition and variations are analyzed with positive matrix factorization (PMF), and the volatility distributions of OA and OA factors are quantified using the mass transfer model (Riipinen et al., 2010) together with the method of Karnezi et al. (2014). The volatility distributions between

ambient OA and BC-containing OA, and the differences between 2017 and 2018 are elucidated.

## 2 Experimental methods

### 2.1 Sampling and instrumentation

All measurements in 2018 were conducted at the urban site of Institute of Atmospheric Physics, Chinese Academy of Sciences (39°58′28″N, 116°22′16″E) from 20 May to 23 June. A detailed description of the sampling site is given in Xu et al.

(2015). Ambient particles larger than 2.5 μm were first filtered out by a $PM_{2.5}$ cyclone. After dried by a nafion dryer, the remaining particles passed through an Aerodyne TD, and then sampled by an HR-AMS and a Cavity Attenuated Phase Shift Single Scattering Albedo monitor (CAPS $PM_{SSA}$, Aerodyne Research Inc.) with a total flow rate of 1.4 L $min^{-1}$. The TD was operated by alternating the bypass line and TD line every 15 min, and the HR-AMS was operated in V-mode with a time resolution of 3 min. The temperatures in heating section of TD were set at 50°C, 120°C and 250°C, corresponding to the



measured temperatures of 50°C, 116°C, 226°C, respectively. In addition, the data during the ramp period of temperature were also analyzed and grouped into four bins, i.e., 127°C, 109°C, 90°C and 70°C. In summer 2017, a TD made by the University of California, Davis (Zhou et al., 2016) coupled with the HR-AMS and SP-AMS were used to measure aerosol volatility from 4 June to 13 June. The temperature settings were 50°C, 100°C, 150°C and 260°C. While the operations of

HR-AMS were the same as those in 2018, the SP-AMS was operated with laser vaporizer only, and thus it only measured refractory BC (rBC) and BC-containing aerosol species. Note that the residence time in the heating section of the TD was 1.9 s and 7.4 s in 2017 and 2018, respectively due to the different flow rates. As a result, the thermograms of aerosol species from the two campaigns cannot be directly compared (Saha et al., 2017;An et al., 2007).

## 2.2 AMS data analysis

The HR-AMS data was analyzed by PIKA V 1.15D (http://cires1.colorado.edu/jimenez-group/ToFAMSResources/ToFSoftware/index.html). The ionization efficiency (IE) and relative ionization efficiencies (RIEs) were calibrated using pure $NH_4NO_3$ and $(NH_4)_2SO_4$ following the standard protocols (Jayne et al., 2000). The RIEs used in this study were 1.4 for sulfate and 4.3 for ammonium, and the default values for organics (1.4), nitrate (1.1) and chloride (1.3). Because aerosol particles were dried and only slightly acidic as indicated by

$NH_4^+_{measured}/NH_4^+_{predicted}$ (0.92 and 0.94 in 2018 and 2017, respectively), we applied a collection efficiency (CE) as function of ammonium nitrate mass fraction to ambient data and a constant CE (0.5) to TD data (Huffman et al., 2009a). The elemental composition of OA was determined with the "Improved-Ambient (I-A)" method (Canagaratna et al., 2015). The data analysis of SP-AMS is similar to that of HR-AMS that was detailed in Wang et al. (2019).

The particle losses through TD were corrected by the comparisons of rBC measured by SP-AMS in 2017 and the aerosolized

NaCl measured by a scanning mobility particle sizer (SMPS, TSI Inc.) in 2018 between bypass line and TD line (Huffman et al., 2008). As shown in Fig. S1, the mass fraction remaining at different TD temperatures was relatively constant at approximately 95% in 2017, and ~90% in 2018, which are close to the values reported in London (Xu et al., 2016) and Shenzhen (Cao et al., 2018). In addition, the periods with low concentrations of aerosol species and OA factors were removed in data analysis due to the large uncertainties in calculating MFR (Table S1).

## 2.3 Source apportionment of OA

The high resolution OA mass spectra of both ambient ($MS_{ambient}$) and the combined ambient and thermally denuded data ($MS_{ambient+TD}$) were analyzed with PMF to resolve potential OA factors (Paatero and Tapper, 1994;Ulbrich et al., 2009). Previous studies showed that the combined thermal denuded and bypass line data can enhance the contrast for different OA compounds and facilitate the separation of OA factors (Huffman et al., 2009a). We found that the HOA spectrum from



4-factor solution showed unrealistically high *m/z* 44 in both MS$_{ambient}$ and MS$_{ambient+TD}$. Therefore, the mass spectrum of HOA resolved from the period with high impacts of vehicle emissions (26 May – 7 June, 2018), and cooking OA (COA) from 5-factor solution were used as constrains in subsequent multilinear engine (ME-2) analysis (Paatero, 1999). Four OA factors were identified including LO-OOA, MO-OOA and two primary factors, HOA and COA. The mass spectra and time series of

the four OA factors are shown in Fig. 1, and the comparisons between MS$_{ambient}$ and MS$_{ambient+TD}$ are shown in Fig. S2. Consistent with previous studies, HOA was well correlated with BC ($r^2 = 0.47$), and COA was correlated with C$_6$H$_{10}$O$^+$ ($r^2 = 0.75$). Comparatively, LO-OOA and MO-OOA were highly correlated with C$_2$H$_3$O$^+$ (*m/z* 43, $r^2 = 0.97$) and secondary inorganic aerosol (SIA, $r^2 = 0.91$), respectively. More diagnostic correlations between OA factors and tracers are shown in Fig. S3. The diurnal patterns of four OA factors were also similar to those previously reported in urban Beijing. For example,

HOA presented a pronounced diurnal cycle with high concentrations at night, and COA showed two pronounced peaks during mealtimes. Comparatively, the diurnal profiles of both LO-OOA and MO-OOA were relatively flat, yet the time series were quite different. While PMF analysis of MS$_{ambient+TD}$ in 2017 identified four OA factors, including HOA, COA, LO-OOA and MO-OOA. On the other hand, that of BC-containing OA resolved a rBC-rich factor, an HOA-rich factor, and two oxygenated OA factors, LO-OOA and MO-OOA. Note that COA was not resolved from BC-containing OA likely due to the

fact that COA and BC were externally mixed (Wang et al., 2019). Compared with HR-AMS, OA factors resolved from the SP-AMS spectra were much less oxidized. The O/C ratios of BC-containing LO-OOA and MO-OOA were 0.26 and 0.60, which were much lower than 0.62 and 1.21 for non-refractory OA. These results suggest that BC-containing OA can be substantially different from the ambient OA. A detailed description of the source apportionment of BC-containing OA is given in Wang et al. (2019).

**2.4 Estimation of OA volatility distribution**

The time-dependent aerosol evaporation in TD was simulated using the dynamic mass transfer model (Riipinen et al., 2010). The inputs of the model include the initial mass concentration, particle size, density calculated using the method of Kuwata et al. (2011), residence time, loss-corrected MFR and corresponding temperatures.

The measured thermograms were fitted using six logarithmically spaced *C*\* bins, and different volatility ranges were chosen

for each factor based on the best fits between the measured and predicted thermograms. The enthalpy of vaporization and the mass accommodation coefficient were also estimated, which can affect the evaporation rate and corresponding volatilities. The combinations of all properties with the smallest error (top 1%) were chosen to calculate the "best estimate" following the methods described in Karnezi et al. (2014).



## 3 Results and discussion

### 3.1 Thermograms of aerosol species

Figure 2 shows the thermograms of NR-PM$_1$ species and OA factors in summer of 2018. Consistent with previous studies, MFRs of all species show decreasing trends as the increase of TD temperature. The total mass concentration of NR-PM$_1$ decreased significantly from 31.0 µg m$^{-3}$ to 2.0 µg m$^{-3}$ with ~7% mass left at 226°C, suggesting the presence of low volatility compounds. MFR varied differently among different aerosol species. Nitrate showed the fastest decreasing rate in thermograms, consistent with the results observed in London (Xu et al., 2016) and Shenzhen (Cao et al., 2018). Although ammonium nitrate is semi-volatile, ~10% nitrate mass was still observed at 226°C. Such a considerable remaining fraction at the highest temperature was also observed in southern China (Cao et al., 2018). A possible explanation is that nitrate measured by HR-AMS also contained less-volatile organic and inorganic nitrates (e.g., metallic nitrate and organic nitrates) during summertime in Beijing in 2018. As shown in Fig. 3, the ratio of NO$^+$ to NO$_2^+$ increased substantially as a function of TD temperature reaching ~5.5 at 116°C, which is much higher than that of pure NH$_4$NO$_3$ observed from the IE calibration (~3.5). This result supports the presence of low-volatility organic nitrates (Ng et al., 2017;Hakkinen et al., 2012). According to the method suggested by Farmer et al. (2010), the mass concentration of organic nitrate was estimated to be 1.3 – 3.0 µg m$^{-3}$ assuming that the ratio of NO$^+$/NO$_2^+$ (R$_{ON}$) of organic nitrates was 5 - 10. Organic nitrates on average accounted for 27 % at R$_{ON}$ =5 (11% at R$_{ON}$ =10) of the total measured nitrates, which was lower than those during summertime in the south of China, but was comparable to those during autumn and spring (Yu et al., 2018). As shown in Fig. 3, nitrogen-containing organic ions (e.g. C$_2$H$_6$N$^+$, CHNO$^+$) showed higher MFR than inorganic NO$^+$ and NO$_2^+$ across different temperatures, supporting the lower volatility of nitrogen-containing compounds than ammonium nitrate.

Chloride showed a moderate decreasing rate with 30% mass left at 226°C, a behavior quite different from pure NH$_4$Cl that completely evaporated at 80°C (Huffman et al., 2009a). This result suggests that a considerable fraction of chloride measured by HR-AMS was also in the form of less volatile chloride salts (e.g., KCl) rather than ammonium chloride. The MFR of sulfate changed slowly before 80°C, and then decreased rapidly to approximately 88% at 226°C for the 2018 campaign. This is different from the behavior in 2017 when MFR started declining above 150 °C (Fig. S4). Such differences are due a large extent to different TD characteristics (e.g., residence time). We noticed the changes in SO$^+$/SO$_3^+$ and SO$_2^+$/SO$_3^+$ ratios after 100°C, suggesting the changes in sulfate composition. One explanation is the presence of organosulfates or other inorganic sulfate salts. As shown in Fig. 3, the MFR of CH$_3$SO$_2^+$, a marker ion for methanesulfonic acid (MSA) (Ge et al., 2012) showed a different thermogram compared to SO$^+$, SO$_2^+$ and SO$_3^+$, supporting the different volatility between sulfate and sulfur-containing organic compounds.



28% of particulate organics evaporated at 50°C, a fraction larger than that observed in Shenzhen (~10%) (Cao et al., 2018), Centreville and Raleigh (Kostenidou et al., 2018;Saha et al., 2017) and Athens (Louvaris et al., 2017), yet similar to the observations in Paris (Paciga et al., 2016), suggesting that OA was overall more volatile in Beijing compare to most other sites. At 226°C, around 10% of the organic mass remained, accounting for ~50% of the total NR-PM$_1$ mass (Fig. 2),
indicating an important role of organics in low volatility compounds. While the contribution of low-volatility OA is close to that in London (Xu et al., 2016), it is much lower than that observed during the SOAR-1 and MILAGRO campaign (Huffman et al., 2009a), which might be due to the differences in sources and composition at different sampling sites besides the different residence time and TD properties.

## 3.2 OA composition and thermograms of OA factors

PMF analysis identified four OA factors in the summers of 2018 and 2017. As shown in Fig. 4, SOA (= LO-OOA+MO-OOA) dominated OA during both periods, on average accounting for 65% and 72% in 2017 and 2018, respectively, consistent with the results from previous studies (Sun et al., 2018;Hu et al., 2016). LO-OOA was the dominant SOA factor, accounting for 39% and 45% of the total OA in 2017 and 2018, respectively, while the contribution of MO-OOA was comparable (~26 - 27%). The differences in POA composition between 2017 and 2018 were also observed. Although the contribution of HOA
was comparable (11% vs. 13%), that of COA decreased from 24% in 2017 to 15% in 2018. These results suggest that OA composition had considerable differences between the two years. As shown in Fig. 2, the MFR of HOA was 0.73 at 50°C and then decreased to 0.1 at 226°C. Half of the HOA mass evaporated at ~70°C ($T_{50}$), which was comparable to that measured during the MILAGRO and SOAR-1 campaigns (Huffman et al., 2009a), but slightly higher than that in Shenzhen (Cao et al., 2018) and Paris ($T_{50}$ = 49-54°C) (Paciga et al., 2016). Although the mass concentration of HOA decreased substantially at
higher TD temperatures, its fraction in OA remained relatively constant (~15%). Such results are consistent with those observed at the NK site (16%) and Detling (19%) (Xu et al., 2016), yet larger than that reported in Shenzhen (Cao et al., 2018). Compared to HOA, COA showed a higher $T_{50}$ (~85°C), but lower than that observed in Shenzhen (Cao et al., 2018) and Paris (Paciga et al., 2016), suggesting a higher volatility for COA in Beijing than other cities. One reason might be due to the different cooking methods generating OA with different volatility. Note that the MFR of COA showed slightly higher
values than HOA in the range of 50°C to 120°C, suggesting that COA contained more compounds with high $C^*$ compared with HOA. This was also supported by the higher fraction of $C^* \geq 10$ µg m$^{-3}$ for HOA (51%) than COA (37%, see section 3.3 for more details).

LO-OOA evaporated 33% at $T$=50°C in Beijing, which is comparable to that in Shenzhen (30%) (Cao et al., 2018) and Paris (Paciga et al., 2016), but higher than that in Centreville (Kostenidou et al., 2018). The concentration of LO-OOA decreased
from 5.7 µg m$^{-3}$ at ambient temperature to 0.15 µg m$^{-3}$ at $T$=226°C, and its contribution to OA also decreased from 45% to 15%, indicating a high volatility of LO-OOA. Comparatively, the MFR of MO-OOA showed the slowest decreasing rate in



thermograms among all OA factors. As a result, the fraction of MO-OOA in OA showed an increasing trend and became the dominant component at 226°C (Fig. 2). Previous studies showed that such non-volatile organic compounds might be associated with humic-like substances (HULIS) (Wu et al., 2009), an important component of fine particles in Beijing (Ma et al., 2018). However, MO-OOA in this study evaporated faster than that at other sites, e.g., ~16% evaporation at $T$=50°C compared with 1 – 10% in Shenzhen (Cao et al., 2018), SOAR-1 and MILAGRO campaigns (Huffman et al., 2009a). These results might suggest that MO-OOA in this study was more volatile that those previously reported at other sites due to different SOA composition and properties. We further checked the thermograms of NR-PM$_1$ species and OA factors at different time periods in a day. As shown in Fig. S5, MO-OOA appeared less volatile at nighttime than daytime, while the diurnal changes of LO-OOA volatility were small. The reasons for the differences in the diurnal variability were likely due to the different volatile organic compounds (VOCs) precursors, formation mechanisms and meteorological conditions between day and night.

The O/C increased as a function of TD temperature varying from 0.68 in ambient air to 1.17 at 226°C (Fig.2). Such a behavior was consistent with that previously observed at other sites (Xu et al., 2016;Cao et al., 2018), suggesting that the OA remaining at higher temperature was more oxidized. This is further supported by the higher MFR of oxygenated ions $C_xH_yO_2^+$ than that of $C_xH_yO^+$ (Fig. S6). Note that O/C and MFR was weakly correlated ($r$<0.21), suggesting that O/C might not be a good proxy to indicate the volatility (Hildebrandt et al., 2010;Xu et al., 2016).

**3.3 Volatility distribution of OA factors**

Figure 5 summarizes the volatility distributions of the total OA and four OA factors, and the predicted thermograms are depicted in Fig. S7. The average $C^*$ at different sites can be directly compared in the same VBS volatility range (Table S2). In summer 2018, the average $C^*$ of OA was 0.55 μg m$^{-3}$ with vaporization enthalpy (ΔH) and mass accommodation coefficient (a$_m$) being 105 KJ mol$^{-1}$ and 0.33, respectively. The compounds with $C^*$=1 μg m$^{-3}$, 10 μg m$^{-3}$ and 100 μg m$^{-3}$ referring to semi-volatile organic compounds (SVOC) (Murphy et al., 2014) contributed 17%, 19% and 28% to the total OA, respectively. Comparatively, low-volatility organic compounds (LVOC) with $C^*$=0.01 μg m$^{-3}$ and 0.1 μg m$^{-3}$ (Murphy et al., 2014) accounted for 11% and 12%, respectively. In addition, OA consisted of ~13% extremely low volatility compounds (ELVOCs with $C^*$≤10$^{-4}$ μg m$^{-3}$), consistent with the remained organic mass fraction at 226°C (9%). The SVOC fraction in Beijing in summer 2018 was overall larger than those reported in Finokalia (30-60%) (Lee et al., 2010), Athens (38%) (Louvaris et al., 2017), Centreville and Raleigh (60%) (Saha et al., 2017), and 39-73% in Mexico City (Cappa and Jimenez, 2010). Such results might suggest relatively higher volatility of OA in summer in Beijing than other sites. Note that the ELVOCs in Beijing in summer 2018 was comparable to that reported in Centreville and Raleigh (14%) (Saha et al., 2017), yet lower than that in Athens (30%) (Louvaris et al., 2017).





The volatility of four OA factors was different. The average volatility of MO-OOA was $C^*$=0.70 µg m$^{-3}$ ($\Delta H = 57$ KJ mol$^{-1}$ and a$_m$=0.31). LVOC on average accounted for 40% of MO-OOA, which is comparable to that in Centreville (44%, $\Delta H$ =89 KJ mol$^{-1}$ and a$_m$=1) during summertime (Kostenidou et al., 2018), yet lower than those observed during summertime in Athens and Paris (Louvaris et al., 2017;Paciga et al., 2016). These results supported a relatively more volatile nature of MO-OOA in Beijing during summertime compared with other cities. Similar to the variation of MFR in thermogram, LO-OOA with an average contribution of LVOC for 30% was more volatile ($C^*$=1.58 µg m$^{-3}$) than MO-OOA. This result suggests that the freshly oxidized SOA in Beijing is quite volatile, and may affect OA concentration substantially via gas-particle partitioning (Kostenidou et al., 2018).

SVOC on average contributed 67% to HOA, which was much higher than that from diesel vehicles (May et al., 2013), and traffic emissions near road (Saha et al., 2018), yet close to that observed in Paris (63%) (Paciga et al., 2016). These results suggest that HOA from vehicle emissions in Beijing was relatively more volatile. One reason is the different types of fuel used for vehicles (Saha et al., 2018). Another reason might be due to the much lower diesel emissions in Beijing city because diesel trucks are only allowed to enter the 6$^{th}$ ring road between 0:00 – 6:00. This is consistent with the lowest MFR for HOA during 0:00 – 6:00 at $T > 100$ °C (Fig. S5). It should be noted that ELVOCs accounted for 13% HOA, which was lower than that in Athens (30%) (Louvaris et al., 2017), but comparable to that in Paris (11-13%) (Paciga et al., 2016). These results indicate that a considerable fraction of HOA was non-volatile although it was considered as one of the most volatile OA factors (Paciga et al., 2016;Cao et al., 2018). The $C^*$ of COA was 0.79 µg m$^{-3}$ ($\Delta H$ =95 KJ mol$^{-1}$ and a$_m$ = 0.39), and LVOC on average accounted for 40%. The average COA volatility was relatively comparable with that of MO-OOA possibly due to the fact that COA was dominated by fatty acids with relatively low volatilities (Mohr et al., 2009). However, compared with previous studies in Athens and Paris, the fraction of LVOC in COA in Beijing was much lower (40% vs. 63 – 75%) (Louvaris et al., 2017;Paciga et al., 2016), suggesting that COA in Beijing contained more volatile compounds likely due to the differences in cooking oils and styles.

### 3.4 Volatility comparisons between ambient OA and BC-containing OA

Figure 4 presents a comparison of aerosol composition between HR-AMS and SP-AMS in summer in 2017. The BC-containing aerosol particles were dominated by OA (57%), which was much higher than that (42%) from HR-AMS measurements, while the contributions of secondary inorganic aerosols (nitrate, sulfate and ammonium) were correspondingly lower (21% vs. 46%). The composition of BC-containing OA was also substantially different from ambient OA. First, cooking OA was not observed in BC-containing OA, suggesting that COA was externally mixed with BC and was unlikely coated on BC. Further support is that the diurnal pattern of BC-containing OA did not present two pronounced COA peaks as ambient OA (Fig. S8). Second, OA coated on BC was much less oxidized compared with those in ambient aerosol (O/C = 0.36 vs. 0.57 on average). As a result, the volatility of BC-containing OA was expected to be different from ambient



OA. The estimated volatility distributions and thermograms of ambient OA and BC-containing OA are presented in Figs. S9 and S10.

As shown in Fig. 6, the average volatility of BC-containing OA was $C^*$= 0.62 µg m$^{-3}$, which is larger than that of ambient OA ($C^*$= 0.38 µg m$^{-3}$). Consistently, a lower fraction of LVOC (41%) was observed for BC-containing OA than ambient OA (46%), indicating that the BC-containing OA was overall more volatile than ambient OA. We noticed that such differences in volatility appeared to contradict with the variations in thermograms, which show that more than 81% of ambient OA was evaporated at $T$=260 °C, while it was only 66% for BC-containing OA (Fig. S4). Such discrepancies can be explained by the lower effective vaporization enthalpy of BC-containing OA (71 vs. 54 KJ mol$^{-1}$). The volatility of BC-containing POA and SOA were also different from those of ambient OA. As shown in Fig. S9, the MFR of BC-containing POA was ubiquitously higher than that of ambient POA across different TD temperatures, and also much higher than ambient POA after excluding the influences of COA. As indicated by the estimated volatility distribution, the average volatility of BC-containing POA was $C^*$= 0.69 µg m$^{-3}$, which was much higher than that of ambient POA ($C^*$= 0.37 µg m$^{-3}$), and the contribution of LVOC was correspondingly lower (43% vs. 45%). In contrast, the BC-containing SOA showed a lower volatility than ambient SOA as indicated by lower $C^*$ (0.30 µg m$^{-3}$ vs. 0.49 µg m$^{-3}$) and fraction of SVOC (52% vs. 57%). These results suggest that the BC-containing POA contains more volatile compounds compared to ambient POA. One reason was likely due to the fact that the BC-containing OA contains refractory primary species which cannot be measured by HR-AMS. Another reason was that some low volatile OA from primary emissions were not coated on BC, for example COA.

We also compared the OA volatility between 2017 and 2018. Due to the different TD properties and residence time, the MFR cannot be directly compared. According to the estimated volatility (Fig. 5), we found that OA in 2018 showed higher fraction of SVOC (63%) compared to that in 2017 (54%). As shown in Fig. S10, COA and LO-OOA in 2018 showed relatively high volatility ($C^*$=0.79 and 1.58 µg m$^{-3}$) compared to that in 2017 ($C^*$=0.30 and 0.24 µg m$^{-3}$), while MO-OOA and HOA were less volatile in 2018. One possible explanation is that the volatility measurements reaching 260°C in 2017 can reflect extra low-volatility compounds information compared to those in 2018. Another reason is the different sampling period with different meteorological parameters and OA compositions resulting in differences in volatility.

The Weather Research and Forecasting/Chemistry (WRF-Chem, version 3.7.1) model was used to simulate the volatility distribution of SOA in the summer of 2017. The detailed physical and chemical scheme have been given in Zhang et al. (2019). As shown in Fig. 7, the compounds with $C^*$=10 µg m$^{-3}$ and 100 µg m$^{-3}$ estimated from the thermogram method contributed 18% and 19% to the total OA, respectively, which was comparable to that simulated by WRF-Chem (35% in total). However, considerable discrepancies in contributions of compounds with relatively small $C^*$ were observed. For example, the fraction of compounds with $C^*$=1 µg m$^{-3}$ estimated by WRF-Chem was 45%, which was much larger than that



from the thermogram method (21%). Comparatively, the compounds with $C^*$=0.001 µg m$^{-3}$, 0.01 µg m$^{-3}$ and 0.1 µg m$^{-3}$ estimated from the thermogram method were correspondingly higher (43% vs. 19%). These results suggest that current WRF-Chem model might underestimate the fraction of low volatility compounds considerably.

## 4 Conclusion and Implications

Aerosol volatility was measured using a TD-AMS system in Beijing in the summer of 2017 and 2018. Our results showed overall higher fractions of SVOC and saturation concentrations for OA in Beijing compared with those in other megacities in Europe and US, suggesting that OA was more volatile in Beijing. In contrast, inorganic nitrate and chloride showed higher MFR in thermograms, suggesting the presence of organic nitrates and metallic salts other than ammonium nitrate and ammonium chloride. The volatility of OA and four OA factors were estimated with a mass transfer model. MO-OOA and

COA showed lower volatility than LO-OOA and HOA with the contributions of LVOC being 39.8% and 40.5%, respectively. Comparatively, LO-OOA and HOA presented higher contributions of SVOC (70 and 67%, respectively). We also compared the volatility of ambient OA with that of BC-containing OA. The results showed that the BC-containing POA showed much higher volatility compared with that of ambient POA ($C^*$ = 0.69 µg m$^{-3}$ vs. 0.37 µg m$^{-3}$), while the volatility of SOA was lower ($C^*$ = 0.30 µg m$^{-3}$ vs. 0.49 µg m$^{-3}$), highlighting the very different aerosol composition and volatility between ambient

OA and BC-containing OA. The results showed also that the OA volatility was also different in between the two years depending on meteorological parameters, different sources or aerosol processing. The volatility distributions of SOA estimated from the measurement in Beijing were compared with those predicted by the WRF-Chem model in the summer of 2017. Compared to the results of WRF-Chem model, the lower fraction of compounds with $C^*$=1 µg m$^{-3}$ (21% vs. 45%) and higher fraction of compounds with $C^*$≤0.1 µg m$^{-3}$ (43% vs. 19%) estimated from thermogram methods suggest that current

WRF-Chem model might underestimate the fraction of low volatility compounds considerably. Therefore, parameterizing the real volatility distributions into the WRF-Chem model is needed in future studies for better simulations of SOA in Beijing.

*Data availability.* The data in this study are available from the authors upon request (sunyele@mail.iap.ac.cn).

*Author contributions.* YS designed the research. WX, CX, JW, XG, QW, JZ, WD, YQ, WZ, and YH conducted the measurements. WX, CX, and JW analyzed the data. EK and SP supported the mass transfer model analysis. JZ and JA

provided WRF-Chem data. YL, JL, PF, ZW, and DW reviewed and commented on the paper. WX and YS wrote the paper.

*Competing interests.* The authors declare that they have no conflict of interest.





***Acknowledgements.*** This work was supported by the National Natural Science Foundation of China (91744207, 41575120, 41571130034), and the National Key Research and Development Program of China (2017YFC0209601). The authors would like to acknowledge Lu Xu at California Institute of Technology, Wenyi Yang at Institute of Atmospheric Physics, Chinese Academy of Sciences, and Thomas Berkemeier at Max Planck Institute for Chemistry for helpful discussions.

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

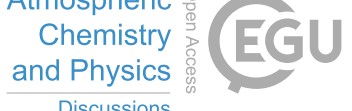

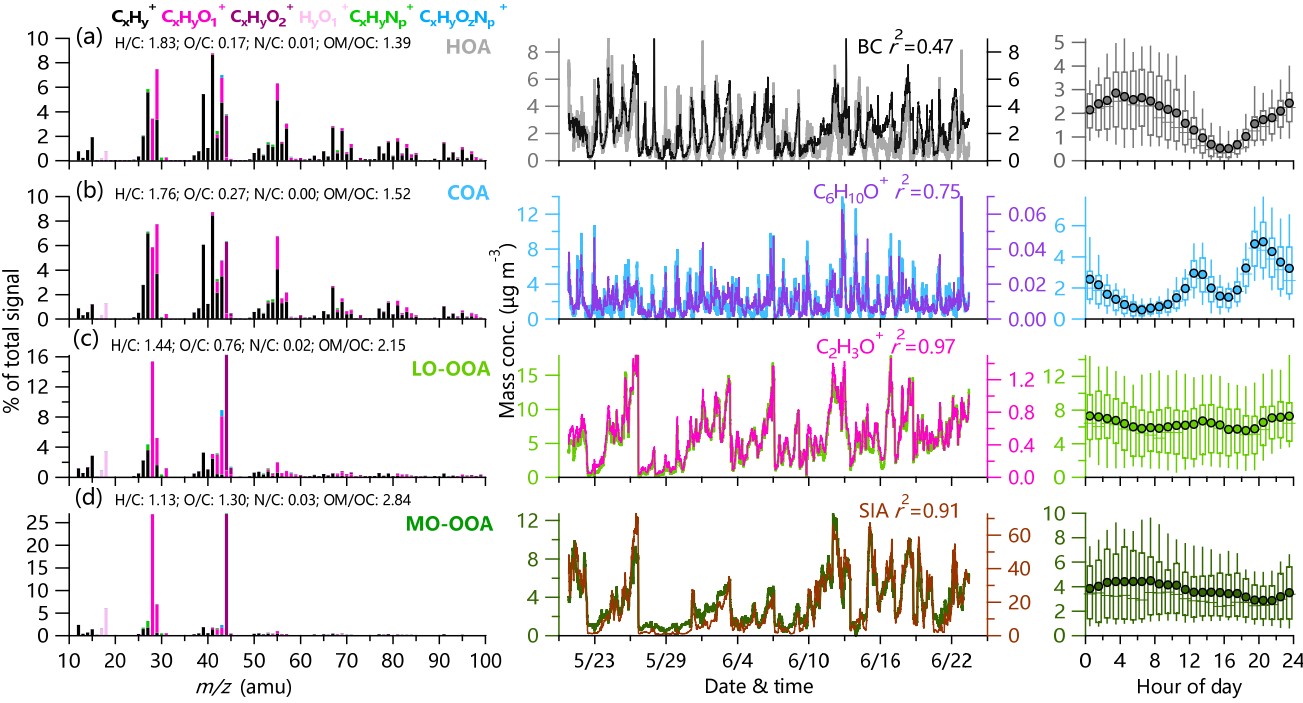

**Figure 1. High-resolution mass spectra (left panel), time series (middle panel), and diurnal patterns (right panel) of four OA factors including (a) HOA, (b) COA, (c) LO-OOA and (d) MO-OOA in summer 2018. Also shown in the middle panels are the time series of other tracers including BC, $C_6H_{10}O^+$, $C_2H_3O^+$, and SIA.**

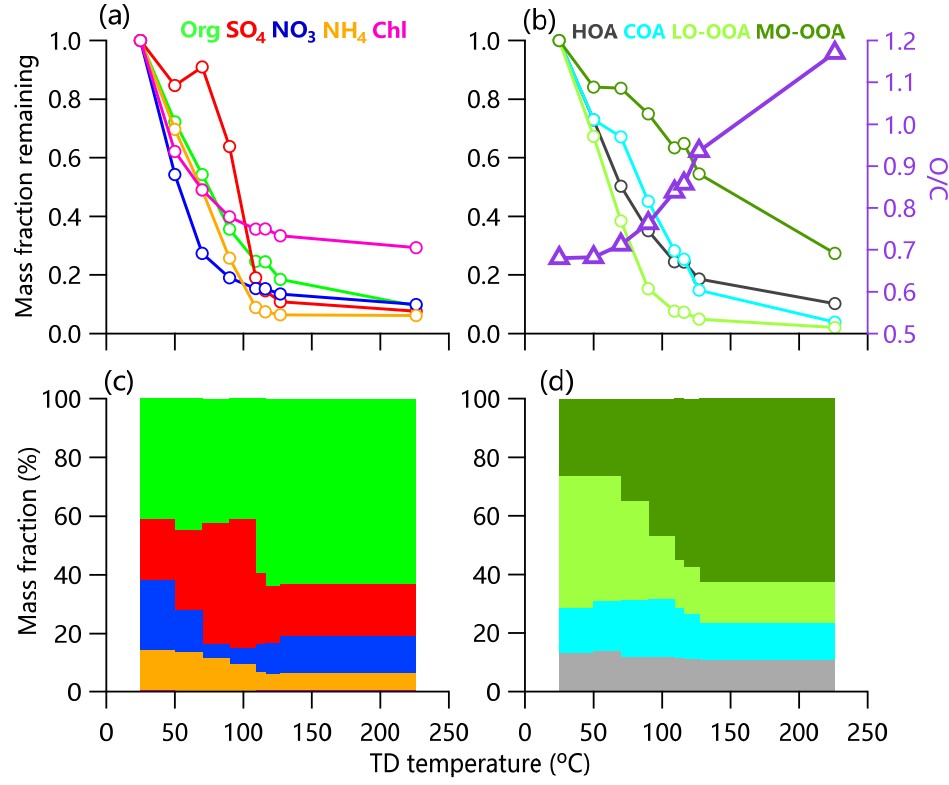

**Figure 2. Thermograms of (a) non-refractory submicron aerosol (NR-PM₁) species (b) OA factors and O/C in summer 2018. (c) and (d) show mass fractions of NR-PM₁ aerosol species and OA factors versus TD temperature.**

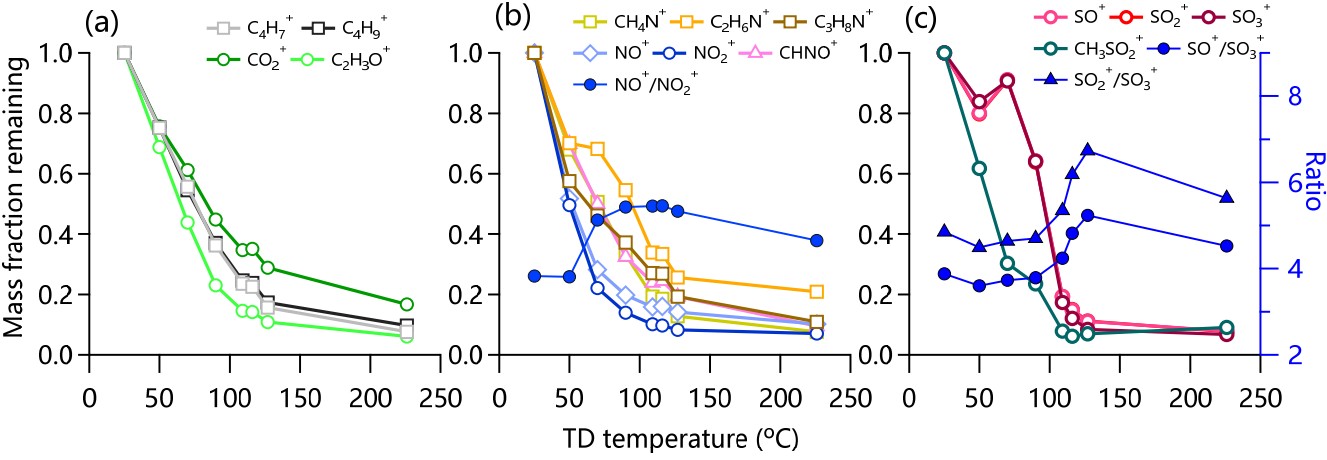





**Figure 3. Thermograms of (a) $C_4H_7^+$, $C_4H_9^+$, $CO_2^+$, $C_2H_3O^+$, (b) $CH_4N^+$, $C_2H_6N^+$, $C_3H_8N^+$, $NO^+$, $NO_2^+$, $CHNO^+$, and (c) $SO^+$, $SO_2^+$, $SO_3^+$, $CH_3SO_2^+$ in summer 2018. The variations of ratios of $NO^+/NO_2^+$, $SO^+/SO_3^+$ and $SO_2^+/SO_3^+$ are shown in (b) and (c).**

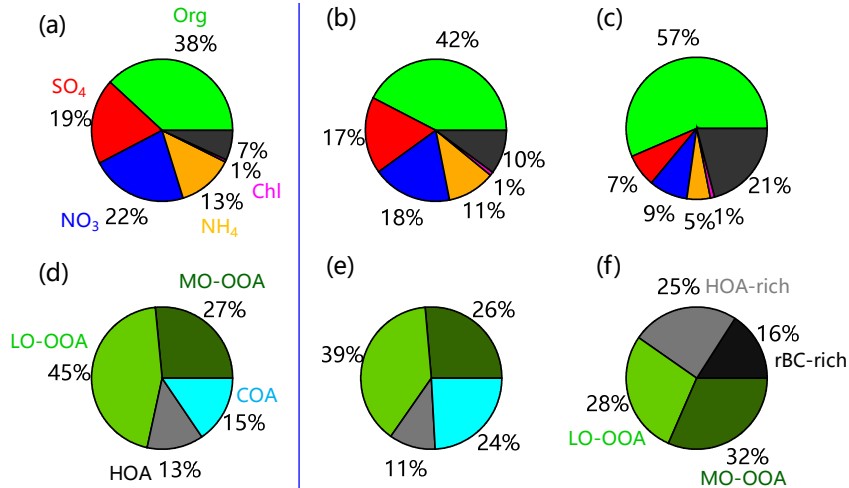

**Figure 4. Average composition of PM$_1$ and OA in summer (a, d) 2018 and (b, e) 2017. The average composition of BC-containing aerosol and OA in summer 2017 is shown in (c, f).**

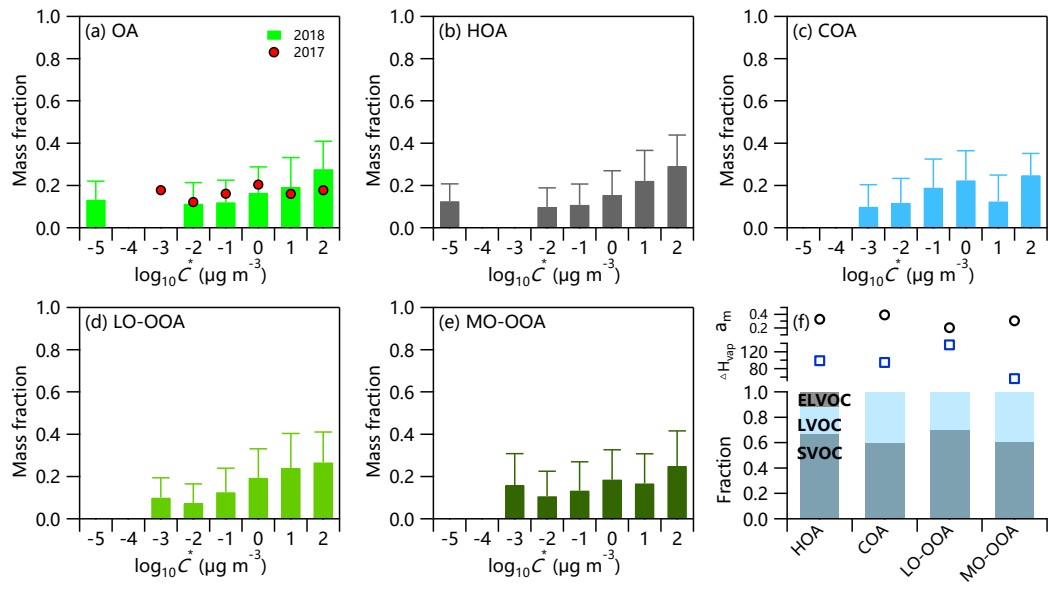



**Figure 5. Predicted volatility distributions of OA and four OA factors in 2018. The error bars are the uncertainties derived using the approach of Karnezi et al. (2014). Vaporization enthalpies, accommodation coefficients, and volatility fractions of SVOC and LVOC for four OA factors are shown in (f).**

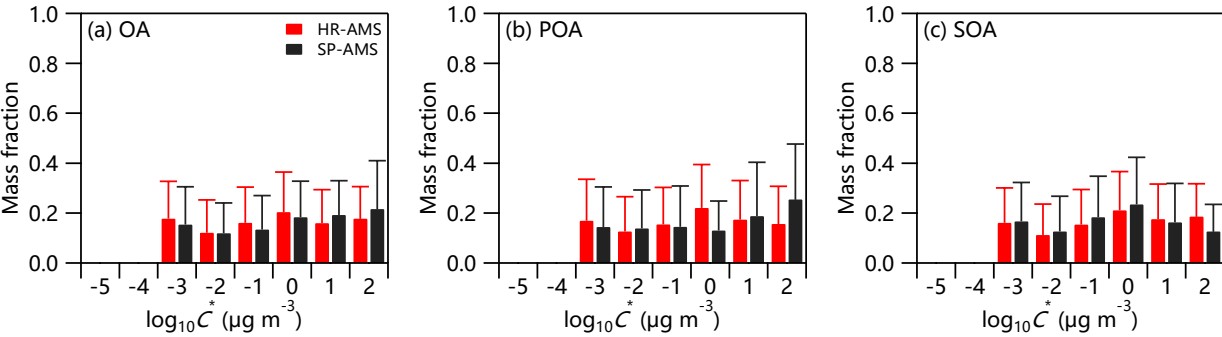

5    **Figure 6. Predicted volatility distributions of OA, POA, and SOA measured by TD-HR-AMS and TD-SP-AMS in 2017. The error bars are the uncertainties derived using the approach of Karnezi et al. (2014).**

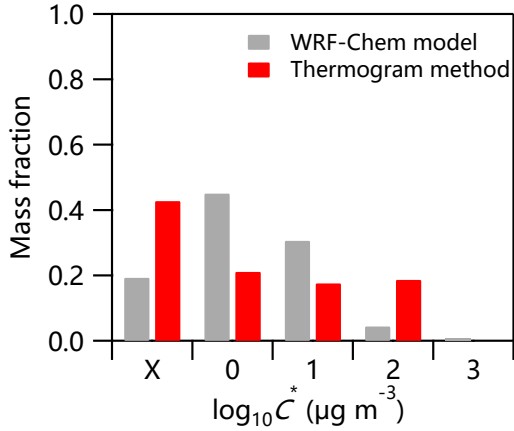

**Figure 7. Volatility distributions of SOA estimated by WRF-Chem model and thermogram method in summer 2017.**