# Peer review of "Summertime aerosol volatility measurements in Beijing, China"

_Atmospheric Chemistry and Physics, 2019_

## Referee Comment (RC1)

**Summertime aerosol volatility measurements in Beijing, China**

W. Xu, C. Xie, E. Karnezi, Q. Zhang, J. Wang, S. N. Pandis, X. Ge, Q. Wang, J. Zhao, W. Du, Y. Qiu, W. Zhou, Y. He, J. Zhang, J. An, Y. Li, J. Li, P. Fu, Z. Wang, D. R. Worsnop, Y. Sun

*Atmos. Chem. Phys. Discuss., https://doi.org/10.5194/acp-2019-135*

**Anonymous referee's comments**

**General comments:**

This manuscript reports results obtained during two field campaigns conducted in Beijing, China, in summer 2017 and 2018. The authors deployed a thermodenuder and a HR-AMS (plus an additional SP-AMS in 2017) to study the volatility distributions of NR-$PM_1$ species, as well as those of OA factors.

This is an important and well conducted study. Even if the manuscript is very descriptive, it will certainly be of interest for the readers of ACP. I would recommend publication of this manuscript after the authors address the following comments.

**Specific comments:**

1) Section 2.1 Sampling and instrumentation: The authors mention that measurements in 2018 were conducted at the Institute of Atmospheric Physics, but nothing is said about the sampling site in 2017. I assume it was at the same location.

2) Section 2.3 Source apportionment of OA: In addition to the description of the PMF analysis, the authors include already here a discussion on the main PMF results. I think that a part of this discussion should be moved into the section 3 Results and discussion.

3) Section 2.3 Source apportionment of OA: did the authors try to perform a PMF analysis by just using TD data from high temperature?

4) Section 3.1 Thermograms of aerosol species: I'm wondering whether the authors identified significant amounts of PAHs in the AMS mass spectra (at least if they measured mass spectra beyond *m/z* 200). These compounds can also have an impact of the volatility of OA factors.

5) Page 6, lines 21-22: Is there a correlation between $K^+$ and $Cl^-$ signals in W-mode? It would also be quite interesting to know the evolution of $NH_4$ measured vs. $NH_4$ predicted as a function of the TD temperature.

6) Page 6, lines 22-29: The thermogram of sulfate for 2018 shows an increase of the MRF at 70-80°C, followed by a fast decrease. Is it possible that a part of sulfate was initially present in particles larger than 1 μm (so not transmitted in the AMS) at ambient temperature, then with

the evaporation, the size of these particles shrank until reaching a size measurable by the instrument? Do the authors have PToF data at different temperatures to check this hypothesis?

7) Page 7, lines 15-16: The authors claim that OA composition had considerable differences between 2017 and 2018. I do not agree with this statement. As shown by the authors just before, the OA composition looks quite similar between the two years.

8) Figure S1: It looks like the legend of the y-axis is wrong. This axis corresponds to the particle loss within the thermodenuder. "Mass fraction remaining" refers to the particle mass which remains after evaporation in the thermodenuder.

9) Figure S6: It does not really make sense to show the thermogram of $C_xH_y^+$ alone in a separate panel. I would suggest including it in the first panel, so that we see the evolution of $C_xH_y^+$, $C_xH_yO^+$, and $C_xH_yO_{gt1}^+$.

---

## Referee Comment (RC2) · Anonymous Referee #2 · 14 Apr 2019

This paper presents aerosol volatility measurements in Beijing using a TD-AMS setup. Measurements are inverted into volatility distributions using an evaporation kinetic model. Volatility distributions are reported for various PMF-resolved OA factors. An improved understanding and characterization of atmospheric aerosol volatility is a topic of interest to many atmospheric researchers. The topic fits well within the scope of ACP. I do have some major comments about presentation, analysis, and discussions in this paper, which should be addressed before acceptance for publication.

Major Comments: 1. Overall, I found the discussions in this paper are limited and incomplete in many cases. It is difficult to identify what are the novel and interesting findings from this study. There are several similar studies exist in the literature. This paper seems another ambient volatility measurement in a different location. Most of

the reported results are also similar to existing studies. It would be nice if the authors can focus a bit more on their novel findings and expand the discussion on it. For example, I found the volatility comparison with WRF-Chem simulation is an interesting part of this paper- since not many studies have done this type of model-measurement comparison. However, the discussion on this comparison is very limited. The authors should consider discussing this result under a separate section. Detail discussion on model-measurement comparisons such as model simulations/inputs, possible reasons for the discrepancy and their implications should be discussed. Also, the implications of their findings in terms of local and regional context should be discussed.

2. It seems the reported volatility distributions may not be well-constrained. They have collected TD data with three temperature steps (50, 120, and 250 degC) with a very low residence time (1.9s in 2017 and 7.4s in 2018). They have used the TD data during the temperature ramp period in their fittings, which seems problematic to me. Because the temperature profile inside the TD during the ramp period may not be in equilibrium. They reported that they had used the fitting method of Karnezi et al. (2014). Details on this should be provided. It is possible to derive hundreds of different volatility distributions by fitting the TD data. The effects of mass accommodation and vaporization enthalpies on the fitted results should be discussed. Ultimately, if their fitted distributions are not well constrained, then all subsequent comparisons among different OA factors and with earlier studies will not be meaningful. In Fig 6, considering the uncertainty, it is difficult to distinguish the difference between the volatility distributions of different OA components.

3. Throughout the paper (especially in Sec. 3.1, 3.2) they have used MFR as a basis for volatility comparison with other studies and/or different OA components in this study. Volatility comparison should not be made based on MFR or T50.

A few specific/minor comments: 1. Page 3, L25: Was the bypass measurement performed after drying? What was RH after drying? Did they characterize and consider particle loss through the dryer?

2. Page 4, L5: Only about one week of data were collected in 2017. Given the different measurement setup and data collection duration, I found a comparison between two-year is a bit problematic. Authors should discuss these limitations. My concern is that they may not be able to resolve the "true difference (if any)" due to measurements limitations and fitting uncertainties and the reported results could be overstated.

3. Page 4, L7: Are the reported residence times plug-flow RT? It should be clarified.

4. A CE of 0.5 is used. Can the author show a mass closure using SMPS measurements (e.g., AMS+BC$\sim$ SMPS)?

5. Page 4, L20: Did they consider size-dependent particle loss in the TD? How do the size distribution of calibration particle (NaCl) and ambient particle compare?

6. Page 5, L18: What fraction of OA was BC-containing OA?

7. Page 5, L22: What particle size information was used for fitting? How did they measure it? Details should be given.

8. Page 7, L10: SOA= LO-OOA+MO-OOA. This may not be always true. They have used SOA and POA in many places, which is sometimes confusing. It is better to use the derived factor.

---

## Author Comment (AC1) · 15 Jun 2019

We are thankful to the two referees for their thoughtful and constructive comments which help improve the manuscript substantially. Following the reviewers' suggestions, we have revised the manuscript accordingly. Listed below are our point-by-point responses in blue to each comment that is repeated in italic.

**Response to Reviewer #1**

**General comments:**

*This manuscript reports results obtained during two field campaigns conducted in Beijing, China, in summer 2017 and 2018. The authors deployed a thermodenuder and a HR-AMS (plus an additional SP-AMS in 2017) to study the volatility distributions of NR-PM1 species, as well as those of OA factors.*

*This is an important and well conducted study. Even if the manuscript is very descriptive, it will certainly be of interest for the readers of ACP. I would recommend publication of this manuscript after the authors address the following comments.*

We thank the reviewer's comments and have revised the manuscript accordingly.

**Specific comments:**

*1) Section 2.1 Sampling and instrumentation: The authors mention that measurements in 2018 were conducted at the Institute of Atmospheric Physics, but nothing is said about the sampling site in 2017. I assume it was at the same location.*

Yes, it was at the same location.

We revised the Section 2.1 as:

"All measurements were conducted at the urban site of Institute of Atmospheric Physics, Chinese Academy of Sciences (39°58'28" N, 116°22'16"E). The TD was operated by alternating the bypass line and TD line every 15 min from 20 May to 23 June in 2018"

2) Section 2.3 Source apportionment of OA: In addition to the description of the PMF analysis, the authors include already here a discussion on the main PMF results. I think that a part of this discussion should be moved into the section 3 Results and discussion.

Thank the reviewer's comments. We have moved the discussions of the PMF results to the section 3.2.

3) Section 2.3 Source apportionment of OA: did the authors try to perform a PMF analysis by just using TD data from high temperature?

We didn't perform PMF analysis with TD data only in this study because we need to calculate MFR for each OA factor. However, PMF analysis of TD data only can introduce large uncertainties in determining the same OA factors due to the large differences in mass spectra with ambient data. Therefore, we used the results from PMF analysis of the combined ambient and TD data to derive

4) Section 3.1 Thermograms of aerosol species: I'm wondering whether the authors identified significant amounts of PAHs in the AMS mass spectra (at least if they measured mass spectra beyond $m/z$ 200). These compounds can also have an impact of the volatility of OA factors.

[Figure]

Figure R1. Average unit mass resolution spectra ($m/z$ 120–350) of OA.

We agree with the reviewer that PAHs can have an influence on the volatility of OA factors. As shown in Figure R1, the average contribution of PAHs to OA was negligible (0.79%) during summertime, which could not affect the volatility of OA in summer significantly.

5) Page 6, lines 21-22: Is there a correlation between $K^+$ and $Cl^-$ signals in W-mode? It would also be quite interesting to know the evolution of $NH_4$ measured vs. $NH_4$ predicted as a function of the TD temperature.

[Figure]

Figure R2. Time series and scatter plot of $K^+$ and Chl measured by HR-AMS and SP-AMS in summer of 2017.

Thank the reviewer's comments. The HR-AMS was only operated in V-mode in this study. Figure R2 shows the correlations between $K^+$ and Chl from SP-AMS and HR-AMS measurements. It can be

seen that Chl was moderately correlated with $K^+$ ($r$ = 0.71 and 0.53), suggesting that a considerable fraction of Chl may exist in the form of KCl.

[Figure]

Figure R3. Variations of average ratio of measured $NH_4^+$ vs. predicted $NH_4^+$ as a function of TD temperature in summer of 2018.

As shown in Fig. R3, the average ratio of measured $NH_4^+$ vs. predicted $NH_4^+$ showed a rapid decrease as a function TD when TD temperature was below 90 °C, consistent with previous studies (Huffman et al., 2009). The reason is that ammonium sulfate is decomposing to yield gas-phase $NH_3$ and acidic $NH_4HSO_4$ at high temperature. However, the average ratio of measured $NH_4^+$ vs. predicted $NH_4^+$ showed a clear increase as temperature increased from 120 °C to 250°C. One possibility is that a large fraction of Chl existed in the form of KCl rather than ammonium chloride. Another explanation is the presence of organic nitrates or other inorganic nitrate salts rather than ammonium nitrate.

6) Page 6, lines 22-29: The thermogram of sulfate for 2018 shows an increase of the MRF at 70-80°C, followed by a fast decrease. Is it possible that a part of sulfate was initially present in particles larger than 1 μm (so not transmitted in the AMS) at ambient temperature, then with the evaporation, the size of these particles shrank until reaching a size measurable by the instrument? Do the authors have PToF data at different temperatures to check this hypothesis?

[Figure]

Figure R4. Size distributions of SO$_4$ for each TD temperature.

Thank the reviewer's comments. It is really a good point. We checked the size distributions of sulfate at different temperatures. As shown in Fig. R4, the peak diameters of sulfate were 650 nm, 600 nm, and 650 nm at TD50, TD70 and TD 90, respectively. Therefore, more transmission from the shrink of particle sizes appears not play a major role for the increases of sulfate at 70 - 80°C. Previous studies showed that such a phenomenon was mainly associated with the decomposition of (NH$_4$)$_2$SO$_4$ into more acidic ammonium bisulfate (NH$_4$HSO$_4$) and gas-phase ammonia (NH$_3$) upon heating, followed by water uptake by the particles after cooling, resulting in particles to form more volatile phases, lessening the bounce off the AMS vaporizer and thus increasing the effective collection efficiency (Larson et al., 1982;Huffman et al., 2009).

7) Page 7, lines 15-16: The authors claim that OA composition had considerable differences between 2017 and 2018. I do not agree with this statement. As shown by the authors just before, the OA composition looks quite similar between the two years.

Thank the reviewer's comments. As shown in Fig.4, the fraction of COA decreased from 24% in 2017 to 15% in 2018, while LO-OOA increased from 39% in 2017 to 45% in 2018. As the reviewer said that "considerable differences" might not be accurate for the description, we deleted it in the revised manuscript.

8) Figure S1: It looks like the legend of the y-axis is wrong. This axis corresponds to the particle loss within the thermodenuder. "Mass fraction remaining" refers to the particle mass which remains after evaporation in the thermodenuder.

Revised

9) Figure S6: It does not really make sense to show the thermogram of $C_xH_y^+$ alone in a separate panel. I would suggest including it in the first panel, so that we see the evolution of $C_xH_y^+$, $C_xH_yO^+$, and $C_xH_yO_{gt1}^+$.

Revised

**Response to Reviewer #2**

This paper presents aerosol volatility measurements in Beijing using a TD-AMS setup. Measurements are inverted into volatility distributions using an evaporation kinetic model. Volatility distributions are reported for various PMF-resolved OA factors. An improved understanding and characterization of atmospheric aerosol volatility is a topic of interest to many atmospheric researchers. The topic fits well within the scope of ACP. I do have some major comments about presentation, analysis, and discussions

in this paper, which should be addressed before acceptance for publication.

We thank the reviewer's comments and have revised the manuscript accordingly.

Major Comments:

1. Overall, I found the discussions in this paper are limited and incomplete in many cases. It is difficult to identify what are the novel and interesting findings from this study. There are several similar studies exist in the literature. This paper seems another ambient volatility measurement in a different location. Most of

the reported results are also similar to existing studies. It would be nice if the authors can focus a bit more on their novel findings and expand the discussion on it. For example, I found the volatility comparison with WRF-Chem simulation is an interesting part of this paper- since not many studies have done this type of model-measurement comparison. However, the discussion on this comparison is very limited. The authors should consider discussing this result under a separate section. Detail discussion on model-measurement comparisons such as model simulations/inputs, possible reasons for the discrepancy and their implications should be discussed. Also, the implications of their findings in terms of local and regional context should be discussed.

Thank the reviewer's comments. Volatility plays an important role in modulating mass concentrations and size distributions of aerosol particles via gas-particle partitioning. Although the TD-AMS has been deployed in various environments including the United States and Europe, few TD-AMS measurements have been reported in polluted regions in China, and the volatility of OA is rarely known. Cao et al. (2018) measured aerosol volatility using TD - AMS in China and discussed the volatility of OA using MFR/T50, which can have large uncertainties in comparing OA volatility in different regions. In this work, we conducted the first OA volatility measurements in two summer seasons in Beijing. Although OA volatility was overall similar to previous studies, we also observed many differences. For example, OA in Beijing comprised mainly semi-volatile organic compounds (SVOC, 63%) and showed overall more volatile properties than OA in megacities of Europe and U.S. In particular, we found that the freshly oxidized secondary OA was the most volatile OA factor rather than the traditional hydrocarbon-like OA (HOA). We also present the first comparison of OA volatilities between ambient bulk composition and BC-containing particles, and found different volatilities of POA and SOA between these two measurements. As the reviewer mentioned, we also compared the OA volatility with that used in WRF-Chem model.

The details on the model simulations and input were given in Zhang et al. (2019) . In WRF-Chem, anthropogenic SOA is produced via the reactions of BIGALK (lumped alkanes C>3), BIGENE (lumped alkenes C> 3), Toluene, Xylenes (lumped isomers of xylene), and Benzene with OH, and biogenic SOA is formed via the reactions of Isoprene, alpha-pinene, beta-pinene and limonene with

OH or $O_3$. SOA mass yields were from Murphy and Pandis (2009) for the volatile organic compound (VOC) precursors and four volatility bins (1, 10, 100, and 1000 μg m$^{-3}$ at 300 K), depending on $NO_x$ conditions and the fraction of organoperoxy radicals that react with NO as opposed to hydroperoxyl and organoperoxy radicals, being the same as those were used in the study of Tsimpidi et al. (2010) and Ahmadov et al. (2012). Aging of VOC oxidation product was considered as Ahmadov et al. (2012). Semi-volatile and intermediate volatility organic compounds are attributed new Henry's law constants (water solubility) calculated from explicit chemistry(Hodzic et al., 2014). These values are included in the calculation of removal through convective and grid-scale precipitation, as well as dry deposition. They strongly affect the removal of SOA as discussed in Knote et al. (2015).

Following the reviewer's suggestions, we expanded the discussions on model simulations and future implications in the revised manuscript.

"One of the major uncertainties in predicting volatility distribution of SOA in WRF-Chem arises from the emission inventories, especially volatile, semi-volatile and intermediate volatility organic compounds. For example, Streets et al. (2003) estimated the overall uncertainty in non-methane VOC (NMVOC) emissions in Asia for the year 2000 to be ±130%, and the uncertainty in NMVOC emissions in China for the year 2005/2006 was in the range of −68% to 120% (Bo et al., 2008;Wei et al., 2008;Zhang et al., 2009;Zheng et al., 2009). Therefore, semi-volatile and intermediate volatility organic compound emissions in China are too limited to be used in SOA simulations (Liu et al., 2017). In addition, model underestimation of atmospheric oxidation capacity, especially in polluted areas, due mainly to the only inclusion of the key gas-phase production of HONO in air quality models (Sarwar et al., 2008;Li et al., 2010;Li et al., 2011;Zhang et al., 2019), and few volatility bins used in WRF-Chem, especially for volatility bin less than 1 μg m$^{-3}$ at 300 K contributed to the discrepancies between model simulation and observations."

"…suggest that current WRF-Chem model might underestimate the fraction of low volatility compounds considerably. Therefore, the uncertainties in emission inventories of VOCs, semi-volatile and intermediate volatility organic compounds need to be reduced substantially to improve the model simulations of OA. Also, more comparisons of model-based and observation-based volatility bins (e.g., 8 or 12 bins) are needed in the future."

2. It seems the reported volatility distributions may not be well-constrained. They have collected TD data with three temperature steps (50, 120, and 250 degC) with a very low residence time (1.9s in 2017 and 7.4s in 2018). They have used the TD data during the temperature ramp period in their fittings, which seems problematic to me. Because the temperature profile inside the TD during the ramp period may not be in equilibrium. They reported that they had used the fitting method of Karnezi et al. (2014). Details on this should be provided. It is possible to derive hundreds of different volatility distributions by fitting the TD data. The effects of mass accommodation and vaporization enthalpies on the fitted results should be discussed. Ultimately, if their fitted distributions are not well constrained, then all subsequent comparisons among different OA factors and with earlier studies will not be meaningful. In Fig 6, considering the uncertainty, it is difficult to distinguish the difference between the volatility distributions of different OA components.

[Figure]

Figure R5. Thermograms of OA factors in summer of 2018. The gray shaded region indicates 95% confidence interval of TD temperature steps (50, 120, and 250 °C). The average values of MFR during the ramping period (crosses) are also shown for a comparison.

According to laboratory measurements, equilibrium can be reached under the condition of high mass concentration larger than 200 μg m$^{-3}$ and the residence time typically needs 30 s (Riipinen et al., 2010;Saleh et al., 2011). Therefore, the ambient TD system is far from equilibrium in both three temperature steps and the ramping period. As shown in R5, the average of data points during ramping periods fall within the 95% confidence interval of three TD temperature steps. Hence, including the TD data during the temperature ramping period appears to be reasonable in this study, and the volatility distributions can be relatively well constrained. In addition, according to the method of Karnezi et al. (2014), we modeled time-dependent evaporation of particles by solving the mass transfer equations, and the residence time has been considered even it is short.

Following the reviewer's suggestions, we expanded the descriptions in section 2.4:

"In order to explore in more detail the solution space, we discretized the parameter space and simulated all combinations of volatilities, $\Delta H_{vap}$ and $a_m$. Briefly, We used logarithmically-spaced effective saturation concentration bins varying the mass fraction of each bin from 0 to 1 with a step of 0.1, the vaporization enthalpy with discrete values of 20, 50, 80, 100, 150 and 200 kJ mol$^{-1}$, and accommodation coefficient with discrete values of 0.01, 0.05,0.1, 0.2, 0.5 and 1. In this case, we derived 96516 different results by fitting the TD data." Hence, the effects of mass accommodation and vaporization enthalpies on the fitted results have been considered.

As shown in Table R1, the absolute values of data were all above 1.962, a threshold value affecting the statistical significance, suggesting that the differences exceeded the 95% significance level for each logarithmically spaced $C^*$ bins in ambient OA/POA/SOA and BC-containing OA/POA/SOA

Table R1. Summary of statistical significance test each logarithmically spaced $C^*$ bins according to the Student t test

| $Log_{10}C^*$ | OA | POA | SOA |
|---|---|---|---|
| -3 | 20.3 | 18.5 | -5.2 |
| -2 | 2.5 | -11.2 | -16.0 |
| -1 | 27.1 | 7.5 | -25.0 |
| 0 | 17.2 | 82.4 | -15.6 |
| 1 | -35.4 | -8.0 | 11.6 |
| 2 | -28.9 | -54.4 | 82.6 |

3. Throughout the paper (especially in Sec. 3.1, 3.2) they have used MFR as a basis for volatility comparison with other studies and/or different OA components in this study.

Volatility comparison should not be made based on MFR or T50.

We totally agree with the reviewer that volatility comparisons should not be made based on MFR or T50, which we claimed in introduction (P2, Line 21). Following the reviewer's suggestions, we have revised the corresponding context in the revised manuscript.

A few specific/minor comments:

1. Page 3, L25: Was the bypass measurement performed after drying? What was RH after drying? Did they characterize and consider particle loss through the dryer?

Yes, aerosol particles were dried with the nafion dryer (MD-700-12, Perma Pure LLC) for both bypass and TD measurements. RH was not measured after the nafion dryer, but expected to be less than 40%. Also, we didn't characterize the particle loss through the dryer. According to the test results from Leibniz Institute for Tropospheric Research and the manufacturer (https://www.permapure.com/wp-content/uploads/2014/06/MD-700-TROPOS-Presentation-10-2014. pdf), the losses are dependent on the size of particles, and as the particle size increases to 40 nm, any losses become insignificant.

2. Page 4, L5: Only about one week of data were collected in 2017. Given the different measurement setup and data collection duration, I found a comparison between two year is a bit problematic. Authors should discuss these limitations. My concern is that they may not be able to resolve the "true difference (if any)" due to measurements limitations and fitting uncertainties and the reported results could be overstated.

We agree with the reviewer that the comparisons between two years may have uncertainties due to different measurement setup and data collection duration. Therefore, we clarified this in section 2.1 "Considering the relatively short time measurements in 2017, the discussions regarding the summer of 2017 focus primarily on the volatility comparisons between ambient OA and BC-containing OA.". In addition, we deleted the volatility comparisons between 2017 and 2018 in the revised manuscript.

3. Page 4, L7: Are the reported residence times plug-flow RT? It should be clarified.

Thank the reviewer's comments. We have revised section 2.1 as below:

"Note that the air residence time (RT) calculated as an average plug flow rate through the heated section of the TD was 1.9 s and 7.4 s in 2017 and 2018, respectively due to the different flow rates."

4. A CE of 0.5 is used. Can the author show a mass closure using SMPS measurements (e.g., AMS+BC_ SMPS)?

[Figure]

Figure R6. Scatter plot of the $PM_1$ vs. $PM_{2.5}$.

Thank the reviewer's comments. The SMPS measurements were not available in this study. We compared the $PM_1$ measured by AMS and AE33 with $PM_{2.5}$. As shown in Fig. R6, $PM_1$ was highly correlated with $PM_{2.5}$ ($r$=0.92), and the average ratio of $PM_1/PM_{2.5}$ was 0.67, consistent with previous studies in Beijing (Sun et al., 2014;Zhao et al., 2017). These results suggest that CE = 0.5 is reasonable for this study.

5. Page 4, L20: Did they consider size-dependent particle loss in the TD? How do the size distribution of calibration particle (NaCl) and ambient particle compare?

[Figure]

Figure R7. Particle loss as a function of size within the TD at three different temperatures.

As shown in Fig. R7, the particle losses increase for smaller sizes and approximately constant above 80 nm. Because the mass contributions of particles smaller than 80 nm (approximately 112 nm in $D_{va}$ assuming a density of 1.4 g cm$^{-3}$) were small (see Fig. R4), the average losses for each experimental curve above 80 nm are used as the integrated number loss over the particle size range where mass is important for ambient particles (Huffman et al., 2008).

6. Page 5, L18: What fraction of OA was BC-containing OA?

The BC-containing OA on average accounted for 49% of the total OA, while the BC-containing secondary inorganic aerosol (sulfate, nitrate, and ammonium) accounted for much less fractions (20 – 25%).

7. Page 5, L22: What particle size information was used for fitting? How did they measure it? Details should be given.

Thank the reviewer's comments. The particle sizes we used for fitting are presented in Table S2. The size distribution of SOA was derived from that of $m/z$ 44 by normalizing the integrated signals of $m/z$ 44 between 30 and 1500 nm to the total concentration of SOA (Zhang et al., 2005). This approach is rationale because SOA was highly correlated with $m/z$ 44 ($R^2$=0.98), while $m/z$ 44 in the mass spectra of POA were generally small. The size distribution of POA was then calculated as the difference between total OA and SOA.

Following the reviewer's suggestions, we expanded the details in the revised manuscript.

8. Page 7, L10: SOA= LO-OOA+MO-OOA. This may not be always true. They have used SOA and POA in many places, which is sometimes confusing. It is better to use the derived factor.

Thank the reviewer's comments. In most cases, the sum of LO-OOA and MO-OOA can be used as a surrogate of SOA. We clarified this in the revised manuscript. It now reads:

"SOA (LO-OOA+MO-OOA as a surrogate)".

References

Ahmadov, R., McKeen, S. A., Robinson, A. L., Bahreini, R., Middlebrook, A. M., de Gouw, J. A., Meagher, J., Hsie, E. Y., Edgerton, E., Shaw, S., and Trainer, M.: A volatility basis set model for summertime secondary organic aerosols over the eastern United States in 2006, J. Geophys. Res., 117, D06301, 10.1029/2011jd016831, 2012.

Bo, Y., Cai, H., and Xie, S. D.: Spatial and temporal variation of historical anthropogenic NMVOCs emission inventories in China, Atmos. Chem. Phys., 8, 7297-7316, 10.5194/acp-8-7297-2008, 2008.

Cao, L. M., Huang, X. F., Li, Y. Y., Hu, M., and He, L. Y.: Volatility measurement of atmospheric submicron aerosols in an urban atmosphere in southern China, Atmos. Chem. Phys., 18, 1729-1743, 10.5194/acp-18-1729-2018, 2018.

Hodzic, A., Aumont, B., Knote, C., Lee-Taylor, J., Madronich, S., and Tyndall, G.: Volatility dependence of Henry's law constants of condensable organics: Application to estimate depositional loss of secondary organic aerosols, Geophys. Res. Lett., 41, 4795-4804, 10.1002/2014gl060649, 2014.

Huffman, J. A., Ziemann, P. J., Jayne, J. T., Worsnop, D. R., and Jimenez, J. L.: Development and Characterization of a Fast-Stepping/Scanning Thermodenuder for Chemically-Resolved Aerosol Volatility Measurements, Aerosol Sci. Tech., 42, 395 - 407, 2008.

Huffman, J. A., Docherty, K. S., Aiken, A. C., Cubison, M. J., Ulbrich, I. M., DeCarlo, P. F., Sueper, D., Jayne, J. T., Worsnop, D. R., Ziemann, P. J., and Jimenez, J. L.: Chemically-resolved aerosol volatility measurements from two megacity field studies, Atmos. Chem. Phys., 9, 7161-7182, 2009.

Karnezi, E., Riipinen, I., and Pandis, S. N.: Measuring the atmospheric organic aerosol volatility distribution: a theoretical analysis, Atmospheric Measurement Techniques, 7, 2953-2965, 10.5194/amt-7-2953-2014, 2014.

Knote, C., Hodzic, A., and Jimenez, J. L.: The effect of dry and wet deposition of condensable vapors on secondary organic aerosols concentrations over the continental US, Atmos. Chem. Phys., 15, 1-18, 10.5194/acp-15-1-2015, 2015.

Larson, T. V., Ahlquist, N. C., Weiss, R. E., Covert, D. S., and Waggoner, A. P.: CHEMICAL SPECIATION OF H2SO4-(NH4)2SO4 PARTICLES USING TEMPERATURE AND HUMIDITY CONTROLLED NEPHELOMETRY, Atmos. Environ., 16, 1587-1590, 10.1016/0004-6981(82)90110-x, 1982.

Li, G., Lei, W., Zavala, M., Volkamer, R., Dusanter, S., Stevens, P., and Molina, L. T.: Impacts of HONO sources on the photochemistry in Mexico City during the MCMA-2006/MILAGO Campaign, Atmos. Chem. Phys., 10, 6551-6567, 10.5194/acp-10-6551-2010, 2010.

Li, Y., An, J., Min, M., Zhang, W., Wang, F., and Xie, P.: Impacts of HONO sources on the air quality in Beijing, Tianjin and Hebei Province of China, Atmos. Environ., In Press, Corrected Proof, 10.1016/j.atmosenv.2011.04.086, 2011.

Liu, H., Man, H., Cui, H., Wang, Y., Deng, F., Wang, Y., Yang, X., Xiao, Q., Zhang, Q., Ding, Y., and He, K.: An updated emission inventory of vehicular VOCs and IVOCs in China, Atmos. Chem. Phys., 17, 12709-12724, 10.5194/acp-17-12709-2017, 2017.

Murphy, B. N., and Pandis, S. N.: Simulating the Formation of Semivolatile Primary and Secondary Organic Aerosol in a Regional Chemical Transport Model, Environ. Sci. Technol., 43, 4722-4728, 10.1021/es803168a, 2009.

Riipinen, I., Pierce, J. R., Donahue, N. M., and Pandis, S. N.: Equilibration time scales of organic aerosol inside thermodenuders: Evaporation kinetics versus thermodynamics, Atmos. Environ., 44, 597-607, 10.1016/j.atmosenv.2009.11.022, 2010.

Saleh, R., Shihadeh, A., and Khlystov, A.: On transport phenomena and equilibration time scales in thermodenuders, Atmospheric Measurement Techniques, 4, 571-581, 10.5194/amt-4-571-2011,

2011.

Sarwar, G., Roselle, S. J., Mathur, R., Appel, W., Dennis, R. L., and Vogel, B.: A comparison of CMAQ HONO predictions with observations from the northeast oxidant and particle study, Atmos. Environ., 42, 5760-5770, 10.1016/j.atmosenv.2007.12.065, 2008.

Streets, D. G., Bond, T. C., Carmichael, G. R., Fernandes, S. D., Fu, Q., He, D., Klimont, Z., Nelson, S. M., Tsai, N. Y., and Wang, M. Q.: An inventory of gaseous and primary aerosol emissions in Asia in the year 2000, J. Geophys. Res.-Atmos., 108(D21), 8809, doi:8810.1029/2002JD003093, 2003.

Sun, Y., Qi, J., Wang, Z., Fu, P., Li, J., Yang, T., and Yi, Y.: Investigation of the sources and evolution processes of severe haze pollution in Beijing in January 2013, Journal of Geophysical Research: Atmospheres, 10.1002/, 2014.

Tsimpidi, A. P., Karydis, V. A., Zavala, M., Lei, W., Molina, L., Ulbrich, I. M., Jimenez, J. L., and Pandis, S. N.: Evaluation of the volatility basis-set approach for the simulation of organic aerosol formation in the Mexico City metropolitan area, Atmos. Chem. Phys., 10, 525-546, 2010.

Wei, W., Wang, S., Chatani, S., Klimont, Z., Cofala, J., and Hao, J.: Emission and speciation of non-methane volatile organic compounds from anthropogenic sources in China, Atmos. Environ., 42, 4976-4988, 10.1016/j.atmosenv.2008.02.044, 2008.

Zhang, J., An, J., Qu, Y., Liu, X., and Chen, Y.: Impacts of potential HONO sources on the concentrations of oxidants and secondary organic aerosols in the Beijing-Tianjin-Hebei region of China, Sci. Total Environ., 647, 836-852, 10.1016/j.scitotenv.2018.08.030, 2019.

Zhang, Q., Streets, D. G., Carmichael, G. R., He, K. B., Huo, H., Kannari, A., Klimont, Z., Park, I. S., Reddy, S., Fu, J. S., Chen, D., Duan, L., Lei, Y., Wang, L. T., and Yao, Z. L.: Asian emissions in 2006 for the NASA INTEX-B mission, Atmos. Chem. Phys., 9, 5131-5153, 10.5194/acp-9-5131-2009, 2009.

Zhao, J., Du, W., Zhang, Y., Wang, Q., Chen, C., Xu, W., Han, T., Wang, Y., Fu, P., Wang, Z., Li, Z., and Sun, Y.: Insights into aerosol chemistry during the 2015 China Victory Day parade: results from simultaneous measurements at ground level and 260 m in Beijing, Atmos. Chem. Phys., 17, 3215-3232, 10.5194/acp-17-3215-2017, 2017.

Zheng, J., Zhang, L., Che, W., Zheng, Z., and Yin, S.: A highly resolved temporal and spatial air pollutant emission inventory for the Pearl River Delta region, China and its uncertainty assessment, Atmos. Environ., 43, 5112-5122, 10.1016/j.atmosenv.2009.04.060, 2009.

---

## Author Response (AR2)

We appreciate reviewer for his/her further comments on our manuscript. Following the reviewer's suggestions, we have revised the manuscript accordingly. Listed below are our responses to reviewer's comments.

In the revised manuscript, the authors have extended some of their discussions. I suggest some minor revisions, after which I recommend publication.

1# Table S2: Please report the particle size information for the bulk OA as well, since the particle size is an important parameter for the evaporation kinetic model. In Table S2, the SOA size diameter peak in 2018 data set is 600 mm and 380 mm in 2017 data set. Any explanation for such big difference in SOA size distribution between two years?

We have added the size information of OA in Table S2.

The average RH in summer of 2017 (from 4 June to 13 June) is 38.0%, which is less than that in summer of 2018 (47.9%), leading to reduction of SOA aqueous processes and consequently reduced particle growth rates. Such effect of RH on the size distributions is consistent with the observation in California (Ge et al., 2012) and Beijing(Hu et al., 2016). In addition, the differences in precursors in two periods also might account for the difference in SOA size distribution.

**2. Please clarify the plug flow residence time at what temperature? I guess it is at room temperature.**

Yes, it is the room temperature.

We have clarified it in revised manuscripts

**3. Page5, L-6: Please provide the C* bin range and rationale for choosing that range.**

Since the OA concentration was on the order 15 μg m$^{-3}$, the thermograms contain little information on the partitioning of compounds with $C^* \geq 1000$ μg m$^{-3}$, in addition, the number of bin was limited by the TD temperatures settings. The different volatility ranges were chosen for each factor based on the best fits between the measured and predicted thermograms. As shown in Fig. S7, the estimates of thermograms were very close to the truth, suggesting that all properties were well estimated.

We have revised section 2.4 as below:

"The measured thermograms were fitted using six logarithmically spaced $C^*$ bins including 100 μg m$^{-3}$, 10 μg m$^{-3}$, 1μg m$^{-3}$, 0.1μg m$^{-3}$, 0.01 μg m$^{-3}$ and 0.001μg m$^{-3}$ (or 0.00001μg m$^{-3}$), and different volatility ranges were chosen for each factor based on the best fits between the measured and predicted thermograms. Since the OA was on the order 15 μg m$^{-3}$, the thermograms contain little information on the partitioning of compounds with $C^* \geq 1000$ μg m$^{-3}$,"

**4. Page 6-7: There are some discussions on volatility comparison between this study and other cities based on TD evaporation data, which are not well supported. I would discourage such comparison. For example, Page-6 "28% of particulate organics evaporated at 50°C, a fraction larger than that observed 5 in Shenzhen (~10%) (Cao et al., 2018), Centreville and Raleigh (Kostenidou et al., 2018;Saha et al., 2017) and Athens (Louvaris et al., 2017), yet similar to the observations in Paris (Paciga et al., 2016), suggesting that OA contained relatively high volatility compounds in Beijing compare to most other sites."**

Since we all know the observed evaporations in TD depend on many experiment specific parameters and thus not a robust measure of volatility, I would suggest to move such discussions in the Sec. 3.3 (Volatility distribution of OA factors) and make comparisons based on the derived volatility distributions.

Thank the reviewer's comments. We have moved the discussion into the section 3.3.

**5. PMF of TD data: I understand the authors performed the PMF for the MS (ambient) and MS (ambient +TD). I assume here MS (ambient +TD) includes all TD data at all the operating TD temperatures. It would be helpful if they can extend the discussion about how they obtained the PMF factors at different TD temperatures.**

Thank the reviewer's comments. We have revised section 2.3 as below:

"Four OA factors were identified including LO-OOA, MO-OOA and two primary factors, HOA and COA. Each factor was separated into ambient data and TD data according to the temperature shift timing recorded by the software of TD"

**6. Figure 7: In my sense, this is a key figure (summary plot) of this paper. I would recommend extending this figure into three panels showing total OA, SOA, and POA from experimental and WRF-simulated results. In addition to OA mass fraction, I would suggest including absolute OA concentrations at different volatility bins, e.g., the left y-axis can show absolute OA concentration (COA), and the right y-axis can show the OA mass fraction.**

Thank the reviewer's comments. It is a good point. However, the WRF-Chem model in this study used the emission inventories where the volatility information of primary OA was not considered. Therefore, we were unable to compare the volatilities of POA and OA. As suggested by the reviewer, such comparisons need to be performed in the future studies.
We have added a figure regarding absolute OA concentration in supplementary.

**7. The volatility of "BC containing POA (C* 0.69 ug/m3)" and "ambient POA (C* = 0.37 ug/m3)" are substantially different. How does the volatility of BC containing POA compare with the volatility of ambient HOA? I think some related discussion would be helpful for the readers.**

Thank the reviewer's comments.
The average volatility of ambient HOA was C*= 0.64 μg m$^{-3}$, which was similar to the BC-coating POA,

We have revised section 3.4 as below:

"Another reason was that some low volatile OA from primary emissions were not coated on BC, for example COA, which was supported by the comparable average volatility between BC-containing POA and the ambient POA after excluding COA (0.69 µg m$^{-3}$ vs. 0.64 µg m$^{-3}$)."

**8. The term "ambient OA" and "BC-containing OA" sounds confusing sometimes. All the measured OA are ambient. I think an appropriate term would be "BC-containing ambient OA". I would suggest to clarify it.**

[revised manuscript text omitted]